# Risk Factors for Development of Canine and Human Osteosarcoma: A Comparative Review

**DOI:** 10.3390/vetsci6020048

**Published:** 2019-05-25

**Authors:** Kelly M. Makielski, Lauren J. Mills, Aaron L. Sarver, Michael S. Henson, Logan G. Spector, Shruthi Naik, Jaime F. Modiano

**Affiliations:** 1Animal Cancer Care and Research Program, University of Minnesota, St. Paul, MN 55108, USA; makie001@umn.edu (K.M.M.); ljmills@umn.edu (L.J.M.); sarver@umn.edu (A.L.S.); henso001@umn.edu (M.S.H.); spect012@umn.edu (L.G.S.); 2Department of Veterinary Clinical Sciences, College of Veterinary Medicine, University of Minnesota, St. Paul, MN 55108, USA; 3Masonic Cancer Center, University of Minnesota, Minneapolis, MN 55455, USA; 4Department of Pediatrics, School of Medicine, University of Minnesota, Minneapolis, MN 55454, USA; 5Institute of Health Informatics, University of Minnesota, Minneapolis, MN 55455, USA; 6Department of Molecular Medicine, Mayo Clinic, Rochester, MN 55905, USA; naik.shruthi@mayo.edu; 7Center for Immunology, University of Minnesota, Minneapolis, MN 55455, USA; 8Stem Cell Institute, University of Minnesota, Minneapolis, MN 55455, USA; 9Institute for Engineering in Medicine, University of Minnesota, Minneapolis, MN 55455, USA

**Keywords:** bone cancer, osteosarcoma, dog, human, pediatric, comparative oncology, genetics, risk factors

## Abstract

Osteosarcoma is the most common primary tumor of bone. Osteosarcomas are rare in humans, but occur more commonly in dogs. A comparative approach to studying osteosarcoma has highlighted many clinical and biologic aspects of the disease that are similar between dogs and humans; however, important species-specific differences are becoming increasingly recognized. In this review, we describe risk factors for the development of osteosarcoma in dogs and humans, including height and body size, genetics, and conditions that increase turnover of bone-forming cells, underscoring the concept that stochastic mutational events associated with cellular replication are likely to be the major molecular drivers of this disease. We also discuss adaptive, cancer-protective traits that have evolved in large, long-lived mammals, and how increasing size and longevity in the absence of natural selection can account for the elevated bone cancer risk in modern domestic dogs.

## 1. Introduction

Osteosarcoma is the most common primary tumor of bone in both dogs and humans. Osteosarcomas are rare in humans, with fewer than 1,000 new cases diagnosed annually [1]. In contrast, osteosarcomas occur more commonly in dogs, although the precise incidence of canine osteosarcoma is not known. A national canine cancer registry does not exist in the United States, European registries are too recent to provide accurate estimates, and most cases of suspected osteosarcoma are not definitively diagnosed histopathologically. Nevertheless, the incidence of osteosarcoma seems to be greater in dogs than in any other species, possibly 10–50 times higher than in humans [1,2,3]. A comparative approach has contributed to our understanding of risk factors associated with osteosarcoma in humans and dogs and has clarified aspects of the disease that are different between the two species. The biological behavior and treatment of osteosarcoma have been the subjects of several recent reviews [4,5,6,7], and the reader is referred to those publications for more extensive coverage of these particular aspects of the disease. The purpose of this review is to highlight risk factors for the development of osteosarcoma in dogs and in humans, underscoring the concept that stochastic mutational events associated with cellular replication are likely to be the major molecular drivers of this disease. 

In both dogs and humans, osteosarcoma has a bimodal age distribution with incidence peaks at two distinct ages (Figure 1). Census-derived population data are available in humans, allowing for the calculation of osteosarcoma rates by age [8]. Comparable data are not available in dogs, and therefore numbers are more commonly presented as frequency among cases. In dogs, the largest incidence peak is observed in older adults [3,9,10], with 80% of cases occurring in animals over 7 years old, and more than 50% of cases occurring in animals over 9 years old. A second, smaller peak comprises approximately 6 to 8% of cases in dogs under 3 years of age (juveniles and young adults). Conversely, peak incidence of osteosarcoma in humans occurs in adolescents and young adults [11,12], with 53% of cases occurring in individuals under 24 years of age. A second, smaller peak occurs in the elderly [1,13], with many osteosarcoma cases in individuals over 60 years of age occurring as a second or later malignancy (24.3%) [1,13], or concomitantly with Paget’s disease (9.5%), which is characterized by abnormal bone turnover in the aging (>55 year old) skeleton [14]. Overall, osteosarcoma is an uncommon complication of Paget’s disease, occurring in an estimated 1 in 650 patients. However, this represents an approximately 30-fold increase compared to the general population >40 years old [15]. 

Osteosarcoma has been reported in every vertebrate class including a wide variety of mammals such as cats [20,21,22,23,24,25,26,27,28,29], horses [30,31,32,33], cows [34,35,36], African hedgehogs [37,38,39], baboons [40,41], guinea pigs [42,43], rabbits [44,45], and others [46,47,48,49], as well as birds [50,51,52,53], fish [54,55], a woma python [56], and two related spiny-tailed monitor lizards [57]. Primary tumors of bone-forming cells (osteoblastomas) have even been identified in dinosaur fossils [58], and osteosarcoma was recently described in a 240-million-year-old fossil of a shell-less stem-turtle [59]. Interestingly, the only published report of osteosarcoma in amphibians is a Russian abstract reporting the disease in an ancient amphibian fossil [60].

Osteosarcoma almost always originates in the skeletal system (Figure 2). It can affect either the appendicular or axial skeleton; however, osteosarcoma is most commonly diagnosed in the long bones of the extremities in both dogs (estimated at 56–86% of cases) [3,9,61] and humans (75.6–96.8% of cases) [1,12,62]. It is worth noting, however, that in species other than dogs and humans, osteosarcoma occurs most commonly in the axial skeleton.

Osteosarcomas of extraskeletal sites, including the skin and subcutaneous tissues [65,66,67,68,69], the liver [65,68,70,71], the lungs [67,72,73,74], and others, have been documented in dogs and humans; however, the frequency of extraskeletal osteosarcoma within all cases of canine osteosarcoma is not known. At a single institution over a 25-year period, Patnaik reported 11 cases of extraskeletal osteosarcoma, compared with approximately 1000 skeletal osteosarcoma cases [75]. The most common site of extraskeletal osteosarcoma documented in two other retrospective studies in dogs was the spleen [65,67], and, in a third study, it was the gastrointestinal tract and the mammary glands [68]. In humans, where there are robust data on osteosarcoma incidence from the Surveillance, Epidemiology, and End Results (SEER) program, extraosseous sites account for fewer than 6% of all osteosarcoma diagnoses, and the frequency increases with age [1]. Less than 1% of osteosarcoma tumors diagnosed in patients between the ages of 0–24 years are extraskeletal, compared with 7% between ages 25–59, and 18% in patients over 60 years old [1]. 

Cancers are caused by DNA mutations that (1) occur as the result of errors during DNA replication or DNA repair, (2) that develop after exposure to environmental carcinogens, or (3) that are inherited in the germline [76]. Recently, Tomasetti and Vogelstein demonstrated a correlation between the risk of cancer development in a particular tissue with the number of stem cell divisions within that tissue [77,78]. With the application of a machine-learning algorithm, they predicted that up to two-thirds of human cancers could potentially result from unavoidable replicative errors. In this model, the molecular etiology of osteosarcoma was almost entirely attributable to replicative errors, rather than environmental exposures or hereditary factors [77]. 

The remaining sections of this review will describe risk factors for the development of osteosarcoma in dogs and humans with particular attention to conditions that increase osteoblast turnover, as each replicative cycle has the potential to contribute to the cellular mutational burden and consequent malignant transformation.

## 2. Hormonal Influence

The peak incidence of age at osteosarcoma diagnosis in children is slightly different in boys (15–19 years) than in girls (10–14 years) [1,2]. This difference in peak age may represent hormonal differences in growth peaks between the two sexes. Males with osteosarcoma (n = 85) began their adolescent growth period and attained their final height earlier than male age-matched control subjects (n = 74) with non-neoplastic orthopedic conditions [79], suggesting that rapid growth could be a risk factor. Earlier onset of puberty was associated with a decreased risk of osteosarcoma in male adolescents in this study [79], suggesting that earlier cessation of growth was protective. 

Sex hormones have known effects on bone density and growth [80,81]. Androgens, such as testosterone, increase longitudinal bone growth and stimulate cellular differentiation into osteoblasts. Estrogens, at higher concentrations, play the predominant role in closure of the growth plate. Testosterone may also contribute to growth plate closure through peripheral aromatization into estrogens, allowing binding to α-estrogen receptors (ERα) at the growth plate. A gender predisposition has been documented in humans, with osteosarcoma occurring more commonly in male adolescents (male:female ratio of 1.34:1) [1,11,12]. Gender predisposition has not been reproducibly documented in canine osteosarcoma; studies have reported a male predisposition (male:female ratio of 1.1:1) [16,17], a female predisposition (male:female ratio of 1:1.1) [18,19], or no significant difference in the proportion of male:female cases [3]. Similarly, associations between reproductive status and development of osteosarcoma have been inconsistent [17,18,19]. Although several reports suggest that spayed and/or neutered dogs have higher incidence of certain cancers [82,83], including osteosarcoma [84], the relationship between reproductive status and cancer risk may be confounded by other variables, such as the documented tendency toward increased adiposity and body condition in gonadectomized dogs [85]. Increased load combined with delayed physeal closure, a result of gonadectomy prior to skeletal maturity [86,87], could theoretically contribute to increased osteosarcoma risk in dogs. A recent, carefully controlled study confirmed a significant association between gonadectomy and longer lifespans in both male and female golden retrievers, as well as a significant difference in age between intact and spayed female golden retrievers that died of cancer, with spayed golden retrievers dying at an older age [82]. This study, which only included cancer cases with confirmed necropsy diagnoses, indicates that the loss of gonadal hormone influence does not increase the risk of cancer, at least in golden retrievers. Rather, the longer lifespan that is associated with gonadectomy might be the major factor that accounted for the previously reported increased risk of cancer in gonadectomized dogs [82]. 

## 3. Size, Height, and Body Weight

The association between appendicular osteosarcoma and size (height and mass) in both dogs [18,19,88,89] and children [12,79] has been shown in multiple studies. Large (25–45 kg) and giant (>45 kg) dogs have an increased risk of osteosarcoma [90] compared with smaller dogs (<10 kg), (odds ratio (OR) 4.2 and 5.6, respectively) [18]. Another study compared standard adult breed height (a variable created as a proxy measure of body height based on published breed standards) and body weight in German shepherd dogs (arbitrarily selected as the reference population) to those in dogs with osteosarcoma. Large (34–44 kg) and giant (>45 kg) breeds had an increased risk of osteosarcoma compared to their reference population (OR 10.3 and 22.8, respectively), as did breeds with increased standard height (>61 cm, OR 15.8) [19].

Similar associations have been well documented in humans with osteosarcoma. A recent meta-analysis evaluating published studies of osteosarcoma cases in humans <40 years old showed that high birth weight (defined as ≥4000 g) was associated with increased osteosarcoma risk [79]. Similarly, a pooled analysis of data from multiple studies showed that, compared with human subjects whose birth weight was between 2665 and 4045 g, individuals with high birth weights (≥4046 g) had an increased osteosarcoma risk [11]. Studies in this pooled analysis included all ages, although the majority of included cases (93.3%) were <24 years old. Increased birth weight was also shown to be associated with more advanced tumor stage at diagnosis [91] and an increased likelihood of the presence of metastatic disease at diagnosis [91,92]. Studies in humans have also shown that taller-than-average adolescents (individuals in the 51st – 89th percentiles of height for their age) and very tall individuals (≥90th percentile) have an increased risk of osteosarcoma (OR of 1.35 and 2.60, respectively) [11,93]. The majority (62%) of human osteosarcoma cases were above the median reference population height for their age. Cases tended to be approximately 0.26 standard deviations taller than the reference population, resulting in an average height difference of 2–3 cm [93].

## 4. Risks from Increased Cell Division/Turnover

The risk of developing osteosarcoma appears to be amplified under conditions that drive excess osteoblast proliferation. Osteosarcoma develops most commonly at or near the site of the growth plates, where cell turnover is highest [94,95]. In addition to overall body size, growth rate might contribute directly to osteosarcoma risk and other abnormalities in skeletal maturation [3,96]. Rapid bone growth results in increased bone remodeling and increased cell turnover. To minimize the risk of developmental bone disease, it is currently recommended to limit energy intake in large breed puppies to prevent rapid growth [96]. Rapid bone growth has been identified as a risk factor for osteosarcoma in humans as well. This is supported by the observations that male adolescents with osteosarcoma tend to have an earlier onset of the adolescent growth period and earlier attainment of final height [79].

Osteosarcoma risk also appears to increase in conditions that drive osteoblast proliferation, such as surgical fracture repair [21,97,98] and orthopedic implant surgery [99,100,101,102,103]. Interestingly, an observation has been stated that AKC-registered “show” greyhounds are at lower risk for developing osteosarcoma than racing greyhounds (G. Couto, unpublished observation, [104]). However, the frequencies of osteosarcoma in racing and show greyhounds have not been published in the peer-reviewed literature, and thus cannot be directly compared. If confirmed, this difference might be due to the effects of concussive forces during training and performance racing, with the potential to create micro-fractures and activate chronic repair processes. Further evidence to support microscopic trauma as a predisposing factor for osteosarcoma is the tendency for appendicular osteosarcoma to occur in weight-bearing bones. Canine appendicular osteosarcoma occurs more commonly in the long bones of forelimbs [19,105], which support approximately 60% of total body weight of the dog. Similarly, in humans, 74.6% of adolescent osteosarcomas occur in the weight-bearing lower limb bones compared with only 11.2% in the bones of the upper limb [1].

There are also anecdotal associations linking osteosarcoma and increased cell turnover due to inflammation. Visceral osteosarcoma secondary to retained surgical sponges has been reported in dogs [65,106], as has a case of osteosarcoma at the site of a previous subcutaneous injection [107]. Several cases of esophageal osteosarcoma are reported in dogs secondary to infection with the nematode parasite *Spirocerca lupi* [108,109,110,111]. There is a case report of orbital osteosarcoma in a cat, several years after enucleation for ocular melanoma, secondary to retention of conjunctival epithelium leading to cyst formation [24]. In humans, osteosarcoma has been diagnosed at the site of cutaneous scars resulting from dermal burn injury [112], following bone graft surgery [113], and at a site of previous electrodessication for actinic keratosis [114]. A case of primary hepatic osteosarcoma was diagnosed in a man with a history of hepatitis C infection and liver cirrhosis [71]. 

## 5. Germline and Somatic Driver Alterations

Unlike many other cancers that are characterized by simple predictable chromosomal rearrangements and relatively low mutation rates, osteosarcoma is a genetically diverse and karyotypically complex cancer [115], characterized by chromosomal instability, copy number alterations, and chromothripsis [116]. The chromosomal instability in osteosarcoma tumors does not necessarily lead to a high mutational burden [117]. There is a moderate mutational burden in osteosarcoma, comparable to many common forms of adult cancers [118], yet recurrent mutations, aside from *TP53*, are infrequent [119,120]. *TP53* was the most commonly mutated gene in a whole exome sequencing (WES) study of matched tumor and normal tissues in dogs, with somatic *TP53* mutations occurring in approximately 60% of tumor samples [121]. This study also found somatic mutations in *SETD2* in 21% of canine tumor samples [121]. *SETD2*, a histone methyltransferase that functions as a tumor suppressor, is mutated in a variety of human cancers [122,123,124,125]; however, somatic *SETD2* mutations occur in <2% of human osteosarcoma samples [126]. 

Osteosarcoma occurs more often in humans with Li-Fraumeni syndrome [127,128,129,130], familial retinoblastoma [131,132,133,134,135,136,137,138,139,140], and Paget’s disease [14,141,142,143,144,145]. It can also occur as a second or later malignancy [131,132,133,134,135,136,137,138,139,140,142]. However, these cases represent a small portion of all osteosarcoma diagnoses, and in approximately 88% of all cases there is no known predisposing condition [1]. The percentage of osteosarcoma cases associated with a predisposing condition varies with age. Only about 4% of osteosarcomas in young human subjects (≤24 years old) occur as a second malignancy or with Paget’s disease, compared with approximately 34% in subjects over 60 years of age [1]. This difference is likely due to the increased prevalence of Paget’s disease and higher likelihood of prior malignancies in older populations. 

Osteosarcoma is one of the most common malignancies seen in patients with Li-Fraumeni syndrome, in which germline *TP53* mutations occur in as many as 83% of cases [146,147]. Adolescent humans with osteosarcoma are more likely to have germline *TP53* variants, with germline *TP53* mutations occurring in 9.5% of young (<30 years old) osteosarcoma patients [148]. Pathogenic germline *TP53* variants were recently reported in 3 out of 95 (3.2%) human osteosarcoma patients <20 years old; importantly, two of the three mutations were considered to be de novo [149].

In hereditary retinoblastoma, an uncommon disorder where people have germline mutations of the retinoblastoma (*RB1*) gene, affected individuals have a reported 69-fold increase in the risk of osteosarcoma, which then increases to greater than 400-fold after radiation therapy [150]. In sporadic (i.e., non-hereditary) cases of osteosarcoma, somatic mutations in the *RB1* gene occur in 20–75% of tumors [119,151,152,153,154]. 

In addition to mutations in *TP53* and *RB1*, a candidate gene association study identified several variants in growth or DNA repair genes that were significantly associated with osteosarcoma [155]. A second candidate gene study of osteosarcoma patients and their unaffected parent(s) discovered variants in estrogen-receptor signaling genes that were significantly associated with osteosarcoma incidence [156]. A genome-wide association study (GWAS) in humans identified two loci associated with the presence of osteosarcoma in a gene desert of chromosome 2, specifically in the glutamate receptor metabotropic 4 (*GRM4*) gene [157]. Additionally, a single, distinct locus in the *NFIB* gene was significantly associated with the presence of metastasis at diagnosis [158]. A trio GWAS study, evaluating pediatric osteosarcoma patients and their unaffected parents, identified genes that achieved significance related to *TP53*, estrogen-receptor signaling, and others [159]. 

GWAS data in three high-risk dog breeds (greyhounds, Rottweilers, and Irish wolfhounds) indicate that the patterns of heritable risk are complex and incompletely penetrant [104]. In that study, 33 loci were enriched in this population of osteosarcoma cases, including variants near *CDKN2A/B*. None of the risk loci were shared between the breeds, but this was due in part to fixed risk alleles for some of the genes in one or more breeds.

Another study, which evaluated whole exome sequencing of matched tumor and normal tissues in three dog breeds predisposed to osteosarcoma (golden retrievers, Rottweilers, and greyhounds), showed germline variants in genes previously associated with osteosarcoma in many of the samples. The most common genes affected were *CDKN2A/B* and *GRM4* in 31.8% and 18.2% of cases, respectively [121]. Additional germline variants commonly seen in human osteosarcoma cases (*RB*, *TP53*, and *NFIB*) were uncommon in this canine cohort, occurring in only one case each [121].

Despite the identification of various germline and somatic mutations in cases of osteosarcoma, we should emphasize that the only recurrent alteration consistently identified in both human and canine osteosarcoma is mutation or copy number alteration of *TP53*. Additionally, it has been shown that the specific alterations affecting *TP53* are not recurrent. 

## 6. Canine Osteosarcoma Provides a Resolution to Peto’s Paradox

In 1977, epidemiologist Richard Peto noted that cancer incidence does not appear to correlate with the size of the organism. He expressed this observation, which came to be known as Peto’s Paradox, by comparing humans and mice: Although humans have 1000 times the number of cells as mice and typically live at least 30 times longer, cancer probability was not vastly different between the two species [160]. This paradox seemingly conflicted with the multistage model of carcinogenesis, proposed in 1954 by Peter Armitage and Richard Doll, which is based on the principle that cancer develops after a cell has accumulated a series of somatic mutations sufficient for carcinogenesis [161]. Based on the multistage model, organisms with increased cellular mass and/or longer lifespans (allowing for more cell divisions and increased exposure to potential DNA damaging agents) should have a higher incidence of cancer development.

Indeed, within a species, there is evidence for a correlation between increased cellular mass and osteosarcoma development. The increased risk of osteosarcoma in large and giant breed dogs, and in children who are in the higher percentiles of size for their age, can be partly explained by the fact that more cell divisions are needed for the formation and continuous remodeling of the long bones of the appendicular skeleton. Somatic mutations occurring during cell division can either undergo DNA repair, cause apoptosis of the cell, or potentially lead to carcinogenesis. Therefore, every replicative cycle of a cell, such as an osteoblast, has the potential to contribute to its mutational burden and transformation. Additionally, appendicular osteosarcoma tends to develop at or near the growth plates of long bones in both dogs and humans, suggesting an association with cell division [94,95]. 

There is also evidence for a correlation between longevity and cancer incidence within a given species. Many cancers are diseases of the aged, likely due to accumulation of a sufficient number or combination of cancer-initiating somatic mutations. The late age of osteosarcoma onset in dogs, although not in humans, is consistent with chronic selection for cells that accumulate a critical complement of mutations.

Therefore, it appears that achieving a large size relatively rapidly is a primary contributor to osteosarcoma risk. Why, then, do “giant” animals not commonly develop osteosarcoma (and other cancers)? If the probability of cancer development were constant across cells, one would expect a higher incidence of cancer in giant animals, such as elephants or whales. On the contrary, cancer incidence in humans and dogs is much higher than in these giant animals. Indeed, there appears to be no significant relationship between body size and cancer risk across different species. A comprehensive survey of necropsy data from captive zoo animals across 36 different mammalian species compared mortality due to any cancer by body size and lifespan [162]. With a minimum of 10 necropsies per species, animals ranged from the striped grass mouse, with a body weight of 51 g and a lifespan of 4.5 years, to the elephant, with a body weight of 4800 kg and a lifespan of 65 years. The authors found no significant relationship between body mass, lifespan, and cancer incidence [162]. The available literature on cancer incidence in large mammals is primarily based on captive animals held in zoos or wildlife sanctuaries; less is known about cancer incidence in wild mammals. Additionally, cancer incidence rates in large mammals typically do not take into account species population size.

Although the multistage model of carcinogenesis appears to remain true within a given species, as larger and longer-lived individuals have higher incidences of cancer, it does not hold true across different species. What, then, is the answer to Peto’s Paradox? It appears that the answer lies in the evolution of anti-cancer protective mechanisms in giant and long-lived species, reducing cancer risk and delaying cancer development beyond reproductive age and for the duration of the lifespan to which particular species have adapted in their unique evolutionary niche. Such mechanisms are unique to each species, conferring sufficient protection for their evolutionary lifespans. 

There is a strong evolutionary pressure to delay cancer development until after the reproductive age in longer-lived animals that tend to reproduce later in life and that invest significant resources of time and energy rearing young. One mechanism that is protective against cancer in larger mammals (>10 kg) is replicative senescence due to telomere shortening. Most human somatic cells cease to express telomerase reverse transcriptase (TERT) after embryonic development leading to decreased telomerase production. With repeated cellular division, the telomeres of most somatic cells shorten. Replicative senescence occurs when telomeres reach a critically short length and the cell enters into a state of cell cycle arrest [163]. While most species with larger body mass have evolved the ability to undergo replicative senescence [164], smaller mammals such as rodents retain expression of telomerase and their cells do not undergo replicative senescence. Somatic mutations [165] and germline polymorphisms [166,167] in genes associated with increasing or maintaining telomere length have been documented in patients with various cancers, including a germline variant in *OBFC1* significantly associated with pediatric osteosarcoma [167], providing further support for the role of replicative senescence in cancer protection. Maintaining or increasing telomere length may increase the risk of cancer development by allowing a cell to undergo more cell divisions, and therefore potentially accumulate more mutations before senescence.

Elephants are the largest living land mammals, with an average body weight of 4800 kg and long lifespans (65 years) [162]; yet, cancer rates in elephants are quite low. The estimated cancer mortality rate in elephants, based on necropsy data from 644 captive African and Asian elephants, is about 5% [162], compared with up to 25% in humans [168]. Based on a mathematical model of carcinogenesis [169], it was determined that a 2.17-fold decrease in mutation rate would be necessary to provide sufficient protection against cancer development in elephants, given that their cellular mass is estimated to be 100 times that of humans [162]. It was recently discovered that in the elephant lineage, the *TP53* gene underwent multiple duplication events after the split from a common ancestor that gave rise to hyraxes and manatees, but that preceded diversification into Proboscideans such as modern elephants and their extinct relatives, mastodons and mammoths (Figure 3) [170]. Humans and elephants each have one copy of *TP53*, but African elephants also have 19 *TP53* duplication events, or retrogenes (*TP53RTG*) in their genome [162,170], 14 of which retain the potential to encode truncated p53 proteins [170]. Similarly, Asian elephants have 12–17 *TP53RTG*, and their extinct relatives, wooly mammoths and Columbian mammoths, also had 14 *TP53RTGs*. Each *TP53RTG* sequence is flanked on both sides by nearly identical transposable elements, which suggests that the copy number expansion was initiated by a single retrotransposition event, followed by repeated rounds of segmental duplication [170]. The expansion of the *TP53RTG* family in the Proboscidean lineage results in an enhanced p53-dependent DNA damage response leading to apoptosis, when compared to smaller relatives [162] such as hyraxes and aardvarks. Moreover, an elephant-specific leukemia inhibitory factor (LIF) pseudogene duplicate, Zombie LIF6, is upregulated by *TP53* in response to DNA damage, leading to apoptosis [171]. This increased *TP53*-dependent apoptotic response in elephants may be protective against cancer development by removing cells with DNA damage from the dividing population before a cancerous state can develop. Despite its potentially protective role in elephants, we should note that *TP53* duplication is neither necessary nor sufficient for cancer protection; other mammals (e.g., whales) have evolved cancer-protective mechanisms distinct from *TP53* duplication, while still others (e.g., rats) have *TP53* duplication, and yet are not known to experience dramatic cancer protection. However, combined with other selective pressures in elephants, amplification of p53 could serve as a protective mechanism for longevity without cancer.

Whales include the largest living animals, yet similar to elephants, they seem to have very low rates of cancer. The bowhead whale is the fifth largest whale species and is capable of living more than 200 years, the longest known lifespan of any mammal. Unlike elephants, the bowhead whale genome does not have extra copies of the *TP53* gene. Rather, these animals have variants and copy number changes in multiple genes that seem to confer protection from cancer and aging, i.e., they are under positive selection [172]. Notably, bowhead whales were found to have increased *ERCC1* (excision repair cross-complementing rodent repair deficiency, complementation group 1), and duplication of *PCNA* (proliferating cell nuclear antigen). Both are involved in DNA repair [172] and represent a distinct, albeit convergent, mechanism to confer cancer protection by decreasing the accumulation of mutations in dividing cells. 

In addition to the large, long-lived species described, there are several smaller long-lived species that are known to have developed unique mechanisms of cancer prevention, such as the naked mole rat, a small rodent that, in captivity, can live more than 30 years. In contrast with other rodents that have high rates of tumor development, there have been only a few instances of cancer reported in the naked mole rat. Proposed cancer-protective mechanisms in this species include a slow cellular duplication rate, which is hypothesized to restrict malignant growth [173], and a reported hypersensitivity to contact inhibition [174]. Additionally, inactivation of either *TP53* or *RB* alone in naked mole rat cells leads to apoptosis, rather than the cellular proliferation seen in other species [174,175]. 

It is apparent that these cancer-protective mechanisms are part of the evolutionary adaptation of each species to its own niche, including lifespan and body size, likely happening over tens or hundreds of millions of years (Figure 3). In addition, despite the fact that only a few of them have been characterized to date, it is likely that such protective mechanisms to prevent or reduce cancer-causing cellular mutations exist in every species, including humans and dogs. Concomitantly, it is likely that these adaptive solutions to enable large size with longevity are unique and specific to the evolutionary history of each species. 

Viewed in this evolutionary context, Peto’s paradox disappears and allows us to conclude that the remarkably high rate of cancers in large and giant breed dogs is a result of artificial selection for large breeds without co-selection for cancer-protective mechanisms that would occur under conditions of truly natural selection. Selective breeding, especially for large size, has enriched risk alleles for osteosarcoma in certain populations that now appear to be fixed (Figure 4) [3,104,176]. The removal of natural selective pressures combined with increased longevity decreases the chances of adaptation across the rest of the genome. Consequently, the risks of mutation associated with normal processes of cell replication during development, growth, and maintenance into adulthood [77,78], are enhanced in large dogs, making it possible to explain their skewed risk of appendicular osteosarcoma by manipulation of their genomic plasticity with extreme selection for size. This explanation is also consistent with the observation that canine osteosarcoma is less common in the axial skeleton (similar to what is seen in other species) than in long bones, accounting for the effects of size and functional/mechanical stresses on bone.

## 7. Conclusions

In summary, the extremely high risk of appendicular osteosarcoma in large and giant breed dogs may often be the result of replicative mutations caused by normal processes of cell division required to create longer bones, with only modest contributions from heritable or environmental factors. Although a comparative approach enhances our understanding of osteosarcoma, we should also acknowledge that in addition to the many similarities between canine and human osteosarcoma, there are important species-specific differences. This recognition is essential to design innovative comparative studies that will maximize the potential for developing safer and more effective strategies to prevent and treat osteosarcoma in both dogs and people. Additionally, insights from large mammals that have evolved cancer-protective mechanisms to adapt to their species-specific niche, including lifespan and body size, can help elucidate distinct undiscovered mechanisms of cancer protection in other species, such as dogs and humans.

## Figures and Tables

**Figure 1 vetsci-06-00048-f001:**
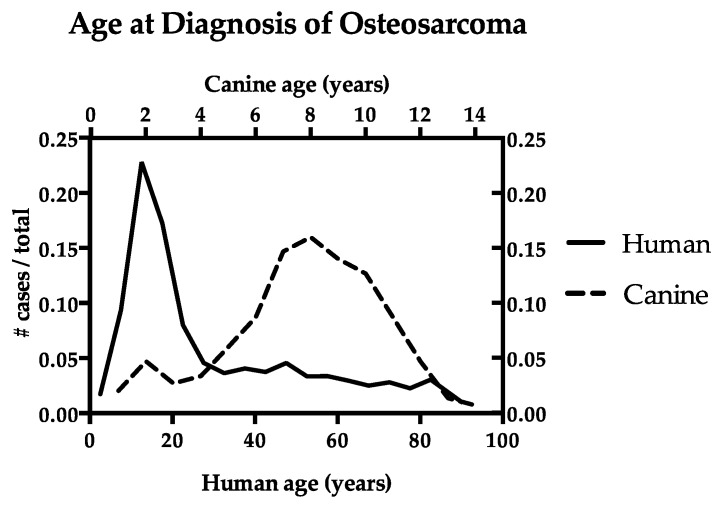
Age at diagnosis of osteosarcoma in dogs and humans. The data are represented as the number of cases diagnosed at each age out of the total number of osteosarcoma cases. Human data (n = 4071) are compiled from the SEER 18 database [8]. Canine cases (n = 150) of histopathologically-confirmed appendicular osteosarcoma, complied from the authors’ database, are consistent with published data [16,17,18,19].

**Figure 2 vetsci-06-00048-f002:**
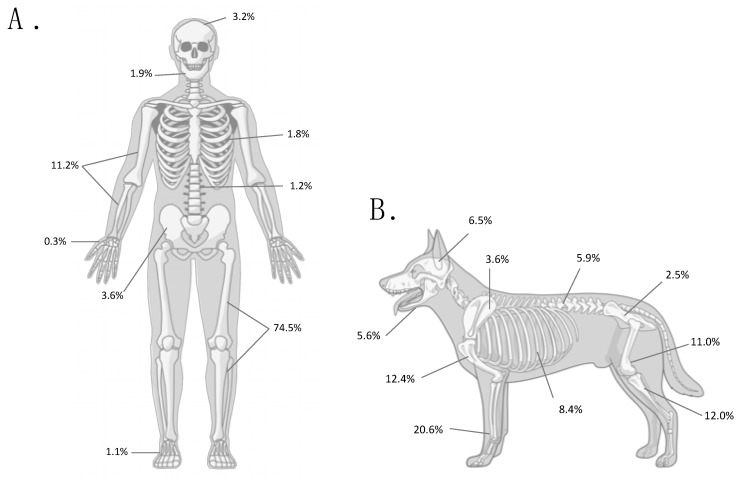
Anatomic distribution of osteosarcoma in humans (**A**) and in dogs (**B**). Numbers indicate the percentage of cases at each anatomic site out of all osteosarcomas in adolescent humans (<24 years old; *n* = 1855) [1], and the mean percentage at each anatomic site compiled from five canine studies (*n* = 1346) [3,9,16,63,64]. Only the most common skeletal sites are included; therefore, percentages do not add up to 100%.

**Figure 3 vetsci-06-00048-f003:**
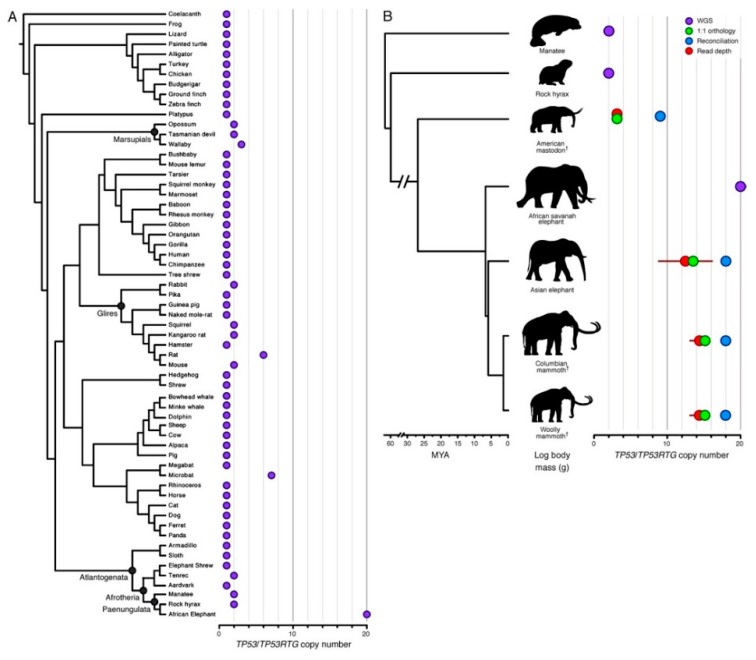
Expansion of the TP53RTG gene repertoire in Proboscideans. (**A**) TP53 copy number in 61 Sarcopterygian (Lobe-finned fish) genomes. Clade names are shown for lineages in which the genome encodes more than one TP53 gene or pseudogene. (**B**) Estimated *TP53/TP53RTG* copy number inferred from complete genome sequencing data (WGS, purple), 1:1 orthology (green), gene tree reconciliation (blue), and normalized read depth from genome sequencing data (red). Whiskers on normalized read depth copy number estimates show the 95% confidence interval of the estimate. Reproduced with permission from “*TP53* copy number expansion is associated with the evolution of increased body size and an enhanced DNA damage response in elephants”, Sulak et al., 2016. https://doi.org/10.7554/eLife.11994.004.

**Figure 4 vetsci-06-00048-f004:**
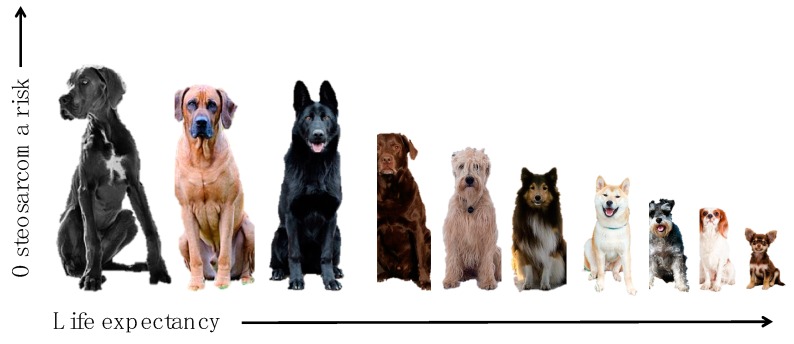
Risk of osteosarcoma and life expectancy are associated with body size in dogs. Large and giant breed dogs generally have shorter lifespans and an increased risk of osteosarcoma compared with smaller breed dogs.

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
