# Peer review of "Risk Factors for Development of Canine and Human Osteosarcoma: A Comparative Review"

_vetsci, 2019, doi:10.3390/vetsci6020048_

Round 1

Reviewer 1 Report

This review presents a summary of risk factors associated with the development of osteosarcoma in dogs and humans. The review is well-organized and comprehensive. While dogs appear to be a relevant, natural model of human osteosarcoma, there are notable differences. Previous reviews put little discussion or importance on these differences. The reviewer appreciates the compare and contrast, where applicable in this summary. The review is also very species comprehensive.  As written, only a couple of items are recommended for transparency and improvement.

Figure 1 is a bold figure with respect to the data not being published. The authors state that the data is determined based upon their own database and compare to what has been published. While certainly not the focus, the authors should comment regarding the thoroughness of their database. Are all of the cases histologically confirmed? Are they all appendicular osteosarcomas? Why not just use what has been published to make the figure if the authors' database is similar to what has been published already?

With the exception of the first figure, the remainder of the figure legends are in a Courier-like text that has uneven spacing and is difficult to read. These should be formatted like figure 1.

Author Response

Dear Reviewer, 

Thank you for your time and consideration in reviewing this manuscript. We have addressed your comments as indicated below. 

Point 1:This review presents a summary of risk factors associated with the development of osteosarcoma in dogs and humans. The review is well-organized and comprehensive. While dogs appear to be a relevant, natural model of human osteosarcoma, there are notable differences. Previous reviews put little discussion or importance on these differences. The reviewer appreciates the compare and contrast, where applicable in this summary. The review is also very species comprehensive.  As written, only a couple of items are recommended for transparency and improvement.

Response 1:Thank you for providing your insight. Indeed, we did try to highlight some of the pertinent differences between canine and human osteosarcoma, as we felt that the similar aspects of the disease had been thoroughly covered in other comparative reviews.

Point 2:Figure 1 is a bold figure with respect to the data not being published. The authors state that the data is determined based upon their own database and compare to what has been published. While certainly not the focus, the authors should comment regarding the thoroughness of their database. Are all of the cases histologically confirmed? Are they all appendicular osteosarcomas? Why not just use what has been published to make the figure if the authors' database is similar to what has been published already?

Response 2:We appreciate this feedback. Efforts were made, both with the original submission and with the revision, to locate published data with detail sufficient enough to compile a graph of age at diagnosis. However, as most publications report general data statistics (e.g. mean/median, range, etc.), rather than the number of cases for each age bracket, published data were not sufficiently detailed enough for assembling a graph of age at diagnosis for canine osteosarcoma. Additional details regarding the authors’ database have been added to more clearly describe the data used to assemble this graph. The figure legend now specifies that the graph depicts histopathologically-confirmed appendicular osteosarcoma cases. 

Point 3:With the exception of the first figure, the remainder of the figure legends are in a Courier-like text that has uneven spacing and is difficult to read. These should be formatted like figure 1.

Response 3: Thank you for bringing this formatting error to our attention. We believe this occurred when formatting the document to a PDF and uploading to the journal submission site. We have addressed the figure legends and they currently appear fixed to the authors; however, we will include a note to the editor to check if this formatting error happens again during the conversion/submission. 

Reviewer 2 Report

This review gives an overview of the risk factors for the development of human and canine osteosarcoma. The subject is quite complicated and a comprehensive analysis is not really simple. I understand the difficulty in going into details of each analyzed point, but the risk is being too superficial, and the final result of the paper is more educational than scientific. Moreover, even if I understand the intent of the authors, I think that the part concerning cancer development in “big animals”, even if interesting,  is not really the focus of the paper. I would point more on the comparative aspects between man and dog, instead, since the comparison between human and canine risk factors are not always clear.

Anyway, these are my major remarks:

lines 31 and 47: ref 5 reports that it is a rare tumor in dogs. Please add at least “more” before commonly

line 51: these figures seem quite high, Fenger reported 10 times and Anfinsen refers to old papers; I would at least cite both figures.

Lines 117-20: sentence not very clear.

Line 226: delete “the” after OSA is…

Lines 284-85: this statement does not hold true in man. It should be commented.

Lines 286-98: could the overall number of animals of a single species also contribute to these figures? i.e.: how many human beings or giant breed dogs are there in the world compared to the overall number of elephants? may this difference (if it does exist) explain the different incidence of OSA in these species? Since the total number of cells in each species that could undergo DNA damage or mutations may be quite different. This should be commented.

Lines 306-377: this part is not related to osteosarcoma, but to cancer development in general. Although interesting and mentioned in the abstract, it is not related to the title of the paper.

Author Response

Dear Reviewer, 

Thank you for your time and consideration in reviewing this manuscript. We have addressed your revision comments as indicated below. 

Point 1:This review gives an overview of the risk factors for the development of human and canine osteosarcoma. The subject is quite complicated and a comprehensive analysis is not really simple. I understand the difficulty in going into details of each analyzed point, but the risk is being too superficial, and the final result of the paper is more educational than scientific. Moreover, even if I understand the intent of the authors, I think that the part concerning cancer development in “big animals”, even if interesting,  is not really the focus of the paper. I would point more on the comparative aspects between man and dog, instead, since the comparison between human and canine risk factors are not always clear.

Response 1:Thank you for providing your insight. Several recent publications have reviewed the comparative aspects of human and canine osteosarcoma, and thoroughly covered the similarities between the two diseases. Therefore, we instead chose to highlight biologically significant differences in this review. As this was commented on favorably by reviewer #1, the authors defer to the editors regarding whether more emphasis should be placed on the comparative similarities in this manuscript.

Point 2:lines 31 and 47: ref 5 reports that it is a rare tumor in dogs. Please add at least “more” before commonly

Response 2:Lines 31 and 47 have been revised to state that osteosarcoma occurs “more commonly” in dogs. 

Point 3:line 51: these figures seem quite high, Fenger reported 10 times and Anfinsen refers to old papers; I would at least cite both figures.

Response 3:The manuscript has been revised to now state “possibly 10 – 50 times higher”, and both the Fenger and Anfinsen references are cited. 

Point 4:Lines 117-20: sentence not very clear.

Response 4:We appreciate this feedback and have revised this sentence in an effort to make it clearer. Please see lines 117 – 120 for this revision. 

Point 5:Line 226: delete “the” after OSA is…

Response 5:Thank you – this has been deleted. 

Point 6:Lines 284-85: this statement does not hold true in man. It should be commented.

Response 6:The manuscript has been revised to address this point. Instead of stating “Cancer is a disease of the aged”, the line now states “Many cancers are diseases of the aged, …” The following sentence was also revised, and now states “The late age of osteosarcoma onset in dogs, although not humans, is consistent with chronic selection of cells … ” 

Point 7:Lines 286-98: could the overall number of animals of a single species also contribute to these figures? i.e.: how many human beings or giant breed dogs are there in the world compared to the overall number of elephants? may this difference (if it does exist) explain the different incidence of OSA in these species? Since the total number of cells in each species that could undergo DNA damage or mutations may be quite different. This should be commented.

Response 7:Thank you for bringing up this interesting point. As this would be difficult to quantify at the population level, the manuscript has been revised to reflect the unknown effect of population size on osteosarcoma incidence. We have added “Additionally, cancer incidence rates in large mammals typically do not take into account species population size.” Please see lines 305 – 306 for this addition. 

Point 8:Lines 306-377: this part is not related to osteosarcoma, but to cancer development in general. Although interesting and mentioned in the abstract, it is not related to the title of the paper.

Response 8:Thank you for the insight provided in this point. Although the review of Peto’s paradox and cancer incidence in large mammals might not appear directly related to osteosarcoma risk in dogs/humans, we have chosen to include it in this manuscript to provide background and context for our conclusion that the high rate of cancers in large breed dogs results from artificial selection for size without co-selection for cancer protective mechanisms. We defer again to the editors for the final decision regarding inclusion of this discussion.